# Primary Gastrointestinal T-Cell Lymphoma and Indolent Lymphoproliferative Disorders: Practical Diagnostic and Treatment Approaches

**DOI:** 10.3390/cancers13225774

**Published:** 2021-11-18

**Authors:** Midori Filiz Nishimura, Yoshito Nishimura, Asami Nishikori, Tadashi Yoshino, Yasuharu Sato

**Affiliations:** 1Department of Pathology, Okayama University Graduate School of Medicine, Dentistry, and Pharmaceutical Sciences, Okayama 700-8558, Japan; p2hq21br@s.okayama-u.ac.jp (M.F.N.); yoshino@md.okayama-u.ac.jp (T.Y.); 2Department of General Medicine, Okayama University Graduate School of Medicine, Dentistry, and Pharmaceutical Sciences, Okayama 700-8558, Japan; nishimura-yoshito@okayama-u.ac.jp; 3Department of Medicine, John A. Burns School of Medicine, University of Hawai’i, Honolulu, HI 96813, USA; 4Division of Pathophysiology, Okayama University Graduate School of Health Sciences, Okayama 700-8558, Japan; asami.kei@s.okayama-u.ac.jp

**Keywords:** primary gastrointestinal T-cell lymphoma, enteropathy-associated T-cell lymphoma, EATL, monomorphic epitheliotropic intestinal T-cell lymphoma, MEITL, indolent T-cell lymphoproliferative disorder, ITLPD-GI, NK-cell enteropathy

## Abstract

**Simple Summary:**

It is challenging for pathologists to diagnose primary gastrointestinal T-cell neoplasms. Besides the rarity of the diseases, the small biopsy material makes it more difficult to differentiate between non-neoplastic inflammation and secondary involvement of extra gastrointestinal lymphoma. Since this group of diseases ranges from aggressive ones with a very poor prognosis to indolent ones that require caution to avoid overtreatment, the impact of the diagnosis on the patient is enormous. Although early treatment of aggressive lymphoma is essential, the treatment strategy is not well established, which is a problem for clinicians. This review provides a cross-sectional comparison of histological findings. Unlike previous reviews, we summarized up-to-date clinically relevant information including the treatment strategies as well as practical differential diagnosis based on thorough literature review.

**Abstract:**

Primary gastrointestinal (GI) T-cell neoplasms are extremely rare heterogeneous disease entities with distinct clinicopathologic features. Given the different prognoses of various disease subtypes, clinicians and pathologists must be aware of the key characteristics of these neoplasms, despite their rarity. The two most common aggressive primary GI T-cell lymphomas are enteropathy-associated T-cell lymphoma and monomorphic epitheliotropic intestinal T-cell lymphoma. In addition, extranodal natural killer (NK)/T-cell lymphoma of the nasal type and anaplastic large cell lymphoma may also occur in the GI tract or involve it secondarily. In the revised 4th World Health Organization classification, indolent T-cell lymphoproliferative disorder of the GI tract has been incorporated as a provisional entity. In this review, we summarize up-to-date clinicopathological features of these disease entities, including the molecular characteristics of primary GI T-cell lymphomas and indolent lymphoproliferative disorders. We focus on the latest treatment approaches, which have not been summarized in existing reviews. Further, we provide a comprehensive review of available literature to address the following questions: How can pathologists discriminate subtypes with different clinical prognoses? How can primary GI neoplasms be distinguished from secondary involvement? How can these neoplasms be distinguished from non-specific inflammatory changes at an early stage?

## 1. Introduction

The gastrointestinal (GI) tract is a common site for extranodal lymphoma involvement. Primary GI lymphomas are predominately of the B-cell lineage, and T-cell neoplasms are rare, accounting for 13–15% of GI lymphomas [1,2,3]. The majority (>90%) of primary GI T-cell neoplasms exhibit aggressive behavior and are associated with short progression-free survival and overall survival. In contrast, an indolent condition termed indolent T-cell lymphoproliferative disorder of the GI tract (ITLPD-GI) has been identified. GI T-cell lymphoma and lymphoproliferative disorders are heterogeneous entities consisting of various subtypes with distinct clinicopathological features and prognoses. Therefore, both clinicians and pathologists must be aware of the distinct characteristics of these lesions to ensure that appropriate care is provided.

The revised 4th World Health Organization (WHO) classification in 2017 broadly classifies primary GI T-cell lymphoma as enteropathy-associated T-cell lymphoma (EATL) and monomorphic epitheliotropic intestinal T-cell lymphoma (MEITL). In the 2001 WHO classification, primary GI T-cell lymphoma with digestive symptoms was initially treated as a separate category, termed enteropathy-type T-cell lymphoma (ETL). In 2008, the entity was renamed EATL and further classified into type I and type II. Type I is associated with celiac disease and has a high incidence in Northern Europe. Conversely, Type II is not associated with celiac disease and has a higher incidence in Asian and Hispanic populations [4]. Type I and type II EATLs differ clinically and morphologically, and also exhibit distinct immunological and genetic features. In the revised 4th WHO classification, EATL types I and II have been revised to EATL and MEITL, respectively [5].

In the revised 4th WHO classification, intestinal T-cell lymphoma, not otherwise specified (ITL, NOS) and ITLPD-GI were newly defined. ITL, NOS is defined as T-cell lymphomas arising in the GI tract that are not otherwise specified as EATL, MEITL, anaplastic large cell lymphoma, or extranodal natural killer (NK)/T-cell lymphoma. The clinical course of ITL, NOS is generally aggressive, and this entity may include cases with insufficient immunohistochemical evaluation and cases of secondary involvement of extra-intestinal lymphoma. Currently, ITL, NOS is considered to be a provisional entity [6]. Extranodal NK/T-cell lymphoma of the nasal type (ENKTCL) and anaplastic large cell lymphoma (ALCL) may also occur in the GI tract or involve it secondarily. Furthermore, NK-cell enteropathy has been reported as a pseudomalignant lesion that is often misdiagnosed as lymphoma.

In this review, we summarize the clinical, histological, and immunophenotypic features; molecular characteristics; and latest treatment approaches for primary GI T-cell lymphoma and lymphoproliferative disorders. We provide a comprehensive review of extant literature to answer the following questions: How can pathologists discriminate subtypes with different clinical prognoses? How can primary GI neoplasms be distinguished from secondary involvement? How can these neoplasms be differentiated from non-specific inflammatory changes at an early stage? This review comprehensively compares and summarizes the histological findings of potential differential diseases and is more practical than previous reviews. It also summarizes the latest treat approaches that have not been summarized in existing reviews.

## 2. Enteropathy-Associated T-Cell Lymphoma

### 2.1. Definition and Epidemiology

EATL is a rare and aggressive intestinal T-cell lymphoma. The geographic distribution of the incidence of EATL is distinct, with a high incidence in Europe and in individuals of northern European descent. A previous report demonstrated that the proportion of EATL in peripheral T-cell lymphoma was 9.1%, 5.8%, and 1.9% in Europe, North America, and Asia, respectively [7]. EATL typically occurs in older individuals (60–70 years), and large studies have reported either an equal sex distribution or slight male predominance (male: female ratio of 1.04–2.8:1) [7,8,9,10,11].

EATL is a complication of celiac disease [12,13], one of the most common genetic disorders that affects approximately 1% of individuals worldwide [14], with a high prevalence in Europe. Celiac disease is characterized by intolerance to dietary gluten that occurs in individuals with HLA-DQ2 or DQ8, haplotypes of human leukocyte antigen (HLA) class II [14]. The association between celiac disease and EATL was first established in a study demonstrating that the HLA risk alleles (HLADQA1*0501 and DQB1*0201 (HLA-DQ2)) of celiac disease are present in majority of patients with EATL [15]. Moreover, serological evidence of gluten sensitivity in patients with EATL has been reported [12]. Notably, a gluten-free diet reduces the risk of lymphoma in patients with celiac disease [16].

### 2.2. Pathogenesis

EATL may be preceded by refractory celiac disease (RCD), which is defined as the persistent or recurrent symptoms of malabsorption and mucosal damage despite a strict gluten-free diet for over 12 months. Exclusion of other etiologies, including autoimmune enteropathy, tropical sprue, and lymphoma, is mandatory to diagnose RCD [17]. RCDs are biologically heterogeneous and can be divided into RCD type I (RCD I) and RCD type II (RCD II) based on immunophenotypic and molecular characteristics of intraepithelial lymphocytes (IELs). RCD I is characterized by increased polyclonal intraepithelial IELs of normal immunophenotype (sCD3^+^, CD8^+^, and CD103^+^), whereas RCD II is characterized by monoclonal IELs of aberrant immunophenotype (sCD3^−^, cCD3^+^, CD8^−^, CD5^−^, and CD103^+^). IELs in celiac disease and RCD I show downregulated expression of CD5, but they are not entirely CD5-negative, and they may have a mixture of CD5-positive and negative subsets in some cases. Although the vast majority of RCD II show negative CD8, some patients who meet the clinical criteria for RCD and show CD8 positivity have been reported to show consistent monoclonality by PCR analysis of the TCR gene throughout the follow-up [18]. Progression of RCD I to RCD II is considered to be rare [19], and the risk of developing EATL is lower in RCD I (3–14% over 5 years) than in RCD II (33–52%) [20,21,22,23,24] (Table 1).

The NK receptor NKp46 was recently reported to be expressed in larger numbers of IELs in RCD II and EATL, whereas only a few IELs were positive in celiac disease and RCD I, suggesting that NKp46 may be a novel biomarker to clarify diagnosis [26].

Currently, two pathways are recognized for the pathogenesis of lymphoma, which are associated with different clinical presentations and outcomes. EATL secondary to RCD II (54% of all EATLs, according to one study [9]) is associated with severe GI symptoms and higher mortality (5-year survival of 0–8%). In contrast, “de novo” EATL (46%), which occurs in patients with uncomplicated celiac disease or RCD I, has higher survival rates (5-year survival of 59%) [9,21]. Nevertheless, the pathogenesis of de novo EATL remains unclear.

### 2.3. Cell Origin

Cells that proliferate in RCD and EATL were formerly thought to be thymic-derived conventional intraepithelial T cells, which is T-cell receptor (TCR) αβ+, due to their immunophenotypic similarity to normal intraepithelial T-cells, which account for approximately 80% of IELs [9,27]. Recent molecular and immunophenotypic analyses suggest that lineage-negative innate IELs are the origin of a proportion of RCD II cases [28,29,30,31], and EATL arising from RCD II may have the same origin.

### 2.4. Histopathology

In EATL, diffuse proliferation of pleomorphic medium to large lymphoma cells is observed. Lymphoma cells have abundant eosinophilic to pale cytoplasm and nuclei that are round or irregular with distinct nucleoli. Infiltration of inflammatory cells, including histiocytes, eosinophils, neutrophils, and plasma cells, is often present in the background [32]. Angiocentric proliferation and vascular invasion are present, and extensive necrosis associated with vascular occlusion is often observed [5,33]. The peripheral intestinal mucosa often exhibits features of celiac disease such as intraepithelial lymphocytosis and villous atrophy.

### 2.5. Immunophenotype and Genetic Alternations

Neoplastic cells are typically CD3^+^ (cytoplasmic), CD5^−^, CD4^−^, CD56^−^, and diffusely positive for cytotoxic granule proteins (TIA-1, granzyme B, and perforin) and intraepithelial homing integrin CD103. CD8 tends to be negative but may be expressed in 19–30% cases [4,9,34] and has been reported to be present at a higher frequency in patients without a history of RCD II [9]. CD30 positivity depends on tumor cell morphology but is almost exclusively positive in large cell-based tumors [9]. Neoplastic cells are negative for anaplastic lymphoma kinase (ALK) and Epstein-Barr virus (EBV). Surface TCR expression is typically absent, but intracellular TCRβ (βF1) expression is observed in approximately 25% of cases [9].

Several studies have used microsatellite markers or array-based approaches to detect recurrent copy number gains at chromosomes 9q (the most common in EATL: 46–70%) [35,36,37], 7q, 1q, and 5q; and losses at chromosomes 16q, 8p, 13q, and 9p. Mutually exclusive gains at 9q and losses at 16q are observed in up to 80% of EATL cases [35,37,38,39]. Moreover, recent studies have reported recurrent mutations of the Janus kinase/signal transducer and activator of transcription (JAK/STAT) pathway with frequent activating mutations in *STAT5B* (26.5–29%), *JAK1* (14.7–23%), *JAK3* (23–27.3%), and *STAT3* (12.1–16%) [36,40].

Targeted next-generation sequencing (NGS) of RCD II cases revealed recurrent activating mutation in *JAK1* (75%) and *STAT3* (25%) genes [30], which may implicate JAK-STAT pathway mutations to be early events in EATL. Targeted NGS analysis of RCD II also revealed frequent occurrence of deleterious mutations in nuclear factor kappa-light chain-enhancer of activated B-cells (NF-κB) regulators and in several epigenetic regulators [41].

### 2.6. Clinical Manifestations

Approximately 90% of EATLs occur in the small intestine [7,8]. Multiple lesions occur in 32–54% of cases, and single lesions in the stomach or colon are rare [7,8,9]. Given the close association between EATL and celiac disease, symptoms such as diarrhea, abdominal pain, weight loss, and hypoalbuminemia may precede EATL [7,9,10,42,43,44]. However, the diagnosis of celiac disease is made prior to the diagnosis of EATL in 20–73% of cases [8,9,42]. EATL also causes vomiting due to intestinal obstruction, intestinal bleeding, and intestinal perforation in 25–50% of patients [7,8,9,43,44,45]. Hemophagocytic syndrome is reported in 16–40% of cases [9,46].

EATLs can also spread beyond the GI tract, and the most common sites of involvement are abdominal lymph nodes (35%), followed by bone marrow (3–18%), lung and mediastinal lymph nodes (5–16%), liver (2–8%), and skin (5%) [7,9,10]. Involvement of the central nervous system (CNS) has also been reported [47,48,49]. Endoscopically, EATL can manifest as a large mass, ulceration, or stricture [7,9].

### 2.7. Prognosis and Treatment Strategies

The prognosis of EATL is very poor. Gale et al. reported 1- and 5- year survival rates of 38.7% and 9.7%, respectively, and 1- and 5-year failure-free survival rates of 19.4% and 3.2%, respectively [8]. Other studies have estimated 5-year survival rates of 11–20% [10,42,43,45,50]. Standard validated treatment strategies for EATL have yet to be established. Surgery and chemotherapy are the treatments of choice, but EATL tends to be refractory to these therapies [50]. CHOP (cyclophosphamide, hydroxydaunorubicin, oncovin, and prednisolone) regimen is the most widely implemented approach, but its overall median survival has been reported to be only 7 months [8,42,43,45]. Recently, autologous stem cell transplant (ASCT) following chemotherapy has been reported to significantly enhance survival [9,10,50,51,52]. One study demonstrated that the novel regimen IVE/MTX (ifosfamide, vincristine, etoposide, methotrexate) followed by ASCT improved survival rates compared to anthracycline-based chemotherapy, with a 5-year overall survival rate of 60% [10]. Although surgery is not considered as first-line treatment for lymphoma, tumor reduction surgery has been reported to be an independent prognostic factor of nutritional status [9]. Further, it is estimated that the combination of surgery and chemotherapy will reduce tumor necrosis, peritonitis, and intestinal hemorrhage associated with chemotherapy, underscoring the potential of surgery as a treatment option at the appropriate time.

One case report of a CD30-positive patient with EATL demonstrated that targeted therapy using brentuximab vedotin resulted in complete remission [53]. In another report, a patient with EATL who had multiple relapses following ASCT achieved sustained remission with CD30 chimeric antigen receptor-modified T-cell therapy [54].

For RCD, various immunosuppressive medications such as azathioprine, systemic corticosteroids, or regular budesonide have been used. Although conventional immunosuppressive medications fail in about half of the cases, recent study revealed open-capsule Budesonide has shown clinical and histological improvement in about 90% of the cases, including those in which conventional treatments have failed [55]. Furthermore, in follow-up, 53% of RCDII patients treated with open-capsule budesonide showed absence of the former clonal TCR gene rearrangement and aberrant IEL phenotype. This indicates that open-capsule budesonide may reduce the risk of developing from RCDII to EATL, although longer follow-up is needed [55].

## 3. Monomorphic Epitheliotropic Intestinal T-Cell Lymphoma (MEITL)

### 3.1. Definition and Epidemiology

Similar to EATL, MEITL is a primary GI T-cell lymphoma deriving from IELs with no apparent association with celiac disease [56]. It occurs worldwide and is more common in Asian and Hispanic populations than in western populations, accounting for the majority of primary GI T-cell lymphomas in Asia [4,7,57,58,59,60]. MEITL is more common in the elderly, with a median age of onset of 58–62 years (range: 23–89 years), and men are more commonly affected than women (male: female ratio of approximately 2:1) [57,58,59].

### 3.2. Histopathology

Similar to EATL, lymphoma cells in MEITL infiltrate extensively into the intestinal mucosal epithelium (Figure 1). MEITL comprises a monomorphic proliferation of relatively small to medium-sized tumor cells with pale cytoplasm, round or slightly irregular nuclei, poorly aggregated nuclear chromatin, and inconspicuous nucleoli. Tumor cell size varies among patients, but cytology within the same tumor is monotonous. Compared to EATL, necrosis and background inflammatory cell infiltration are less prominent.

The central zone (CZ) in the center of the tumor constitutes a site of destructive growth of lymphoma cells in the intestinal wall. In the CZ, tumor cells often infiltrate the entire GI wall, resulting in ulceration and perforation. The peripheral zone (PZ) is characterized by lateral extension of tumor cells in the mucosa and submucosa, with less extensive invasion of tumor cells into the muscularis propria than the CZ (Figure 2). Intraepithelial lymphocytosis comprising >20 IELs per 100 epithelial cells, is observed further around the PZ but is indistinguishable from normal mucosa at low magnification in the IEL zone. The IEL zone may be located distant to the edge of the tumor, occasionally >10 cm away [59].

### 3.3. Immunophenotype and Genetic Alternations

Similar to EATL, tumor cells in MEITL are typically CD3^+^, CD5^−^, CD7^+^, CD4^−^, TIA-1^+^, granzyme B^+^, perforin^+^, and CD103^+^. In contrast, most MEITL cells are CD8^+^ and CD56^+^, unlike EATL (Figure 1). Aberrant CD20 expression has been reported in 11–24% of cases, underscoring the need for caution in diagnosis [58,59]. CD30 is generally negative [59]. Megakaryocyte-associated tyrosine kinase (MATK) is positive in 87% of tumor cells in MEITL, and the extent of MATK expression has been reported to be useful for differentiating MEITL from EATL [58,61].

MEITL is heterogeneous in terms of TCR expression, which can be of γδ or αβ derivation. A study from Hong Kong reported that 78% of MEITLs were of γδ origin [59]. Among the reported cases, 6–33% lacked TCR expression (TCR silent), and dual expression of TCR γ and β chains was observed in 16% of MEITL cases [57,58,59,62]. Unlike EATL, there is no association with HLADQA1*0501 and DQB1*0201 genotypes. Gains at 9q are frequently observed in 70–80% of cases [37,62,63], and gains at the 8q24 locus (resulting in MYC amplification) are also commonly observed (29–73%) [37,58]. Gains at 1q and 5q are less common in MEITL than in EATL [37,64]. Further, alterations in *SETD2* (a tumor suppressor that encodes a lysine N-methyltransferase required for histone H3 lysine 36 trimethylation (H3K36me3)) are frequently observed (93–100% of cases), primarily with loss-of-function mutations and/or loss of the corresponding locus [65,66]. As in EATL, activating mutations in components of the JAK-STAT pathway are frequently observed (76–83% of cases), with higher frequencies of mutations in *JAK3* and *STAT5B* reported in MEITL than in EATL [36,40,62,65].

### 3.4. Clinical Manifestations

The most common site of involvement in MEITL is the small intestine, especially the jejunum [4,58,67]. The stomach, duodenum, and large intestine may also be affected, with reported rates of 2.4–12.0%, 31%, and 8.3–23%, respectively [57,67,68]. Symptoms include abdominal pain, diarrhea, weight loss, and GI hemorrhage [57,59]. Since MEITL is not associated with celiac disease, there is typically no history of malabsorption, and it is often detected after acute abdominal symptoms caused by intestinal obstruction or perforation.

Endoscopically, MEITL appears as a single or multiple masses, deep ulcers, superficial ulcers, or relatively normal findings. Mass formation and superficial ulceration are most frequently seen, in approximately 40% of cases [67].

### 3.5. Prognosis and Treatment Strategies

At the time of diagnosis, 31.6–33% of patients are in advanced Lugano stages [69] III–IV and 23–24% of patients are in Ann-Arbor stages III–IV [57,58,70]. In a recent report, median overall survival was 14.8 months (range: 2.4–27.2 months) [70], and 1-, 3-, and 5-year survival rates were reported to be 36–57%, 26–32%, and 32%, respectively [57,58,59,68,70].

Standardized treatment strategies for MEITL have yet to be established. Similar to EATL, CHOP therapy is widely adopted, but the response to treatment is poor. One study reported that the complete remission (CR) rate for CHOP therapy was 37%, while the CR rate for patients receiving other chemotherapies was 71% (*p* = 0.095) [70]. In this report, non-CHOP chemotherapies included CHOEP (CHOP plus etoposide), ICE (ifosfamide, carboplatin, and etoposide), IMVP-16 (ifosfamide, methotrexate, etoposide, and prednisone), EPOCH (etoposide, prednisolone, vincristine, cyclophosphamide, and doxorubicin), and ESHAP (etoposide, methylprednisolone, (etoposide, methylprednisolone, cytarabine, and cisplatin) [70]. Tse et al. reported that the CR rate was higher for L-asparaginase-based regimens (60%) than for CHOP or anthracycline-based regimens (35%) [57]. Liu et al. reported two cases of MEITL treated with chidamide (a new histone deacetylase inhibitor) combined with chemotherapy with slightly improved survival time [71].

Among patients receiving ASCT (up-front and salvage), the 1- and 5-year overall survival (OS) rates were 100% and 28%, respectively, which improved the prognosis [70]. However, these outcomes were unfavorable relative to the improvement in survival rates in EATL reported in studies in Europe. Patients frequently experience local relapse in the GI tract, and relapse in the CNS is occasionally observed (approximately 10% of cases) [70,72]. Although initial CNS prophylaxis is not generally recommended in patients with peripheral T-cell lymphomas (PTCL), it may be beneficial in MEITL. Achieving CR and concurrent ASCT may be essential for improving prognosis in patients with MEITL. However, many patients are unable to tolerate the toxicity of treatment, and further studies are warranted to determine the appropriate induction therapy for MEITL.

## 4. Intestinal T-Cell Lymphoma, Not Otherwise Specified (ITL, NOS)

### 4.1. Definition and Epidemiology

ITL, NOS is a group of aggressive primary GI T-cell lymphomas that do not meet any of the diagnostic criteria for EATL, MEITL, ENKTCL, or ALCL [5]. This group may include entities for which the diagnosis cannot be confirmed due to insufficient immunostaining, small biopsy specimens, or secondary involvement of extra-intestinal lymphoma. Therefore, at present, it is considered a provisional entity [6]. This group may be reorganized into specific disease categories of intestinal T-cell lymphoma as further findings in the clinicopathologic and genetic spectrum become available [33]. With only a few epidemiological studies specifically investigating this category, data on epidemiology are limited [2,6,60,73]. Despite the small number of cases, reports indicate that the mean age of patients is 44 years and the proportion of patients older than 60 years is 21–32.4%. This entity is more prevalent in men [2,60] and in Asia [60,73,74].

### 4.2. Immunophenotype and Genetic Alternations

As ITL, NOS is not considered a specific disease entity, histology and immunostaining findings are not uniform. Reports indicate that the tumor cells have a medium to large size and are frequently pleomorphic [60]. CD4^+^ or CD4^−^/CD8^−^ double-negative phenotypes are common [6,60]. CD8 and CD56 expression are low compared to that in MEITL [60]. EBV-positive cases are also noted in approximately 8% of cases [60]. TIA-1-positive cases are common (92%), but the reported expression of granzyme B (42%) and CD30 (29%) varies among studies [60]. TCR is silent or expresses TCRβ (βF1) in most cases [6,60]. Data regarding genetic alterations in ITL, NOS are limited.

### 4.3. Clinical Manifestations

ITL, NOS can occur in any of the following regions: stomach (40%), small intestine (20–38.2), ileum (20%), and colon (14.7–60%) [2,6]. No clear association with celiac disease or GI symptoms has been reported to date [2,6]. Patients often present with extensive extra-intestinal diseases at the time of diagnosis, but this may include lymphoid tumors that are not primary to the intestine [2,60]. Clinical behavior tends to be aggressive, but several reports have demonstrated better prognosis compared to EATL and MEITL (median overall survival of 35 months) [2,60].

## 5. Indolent T-Cell Lymphoproliferative Disorder of the Gastrointestinal Tract (ITLPD-GI)

### 5.1. Definition and Epidemiology

ITLPD-GI is a newly defined provisional entity in the revised 4th edition of the WHO classification [5]. The literature on low-grade T-cell LPD of the GI tract has been limited to small case series and case reports [75,76,77,78,79,80,81,82,83,84,85,86,87,88,89,90,91,92,93,94,95,96,97,98,99]. The age distribution of patients is 15–77 years, and it is slightly more common in men than in women [84,85,87,90,100]. There are no known ethnic or genetic risk factors. Several reports have suggested an association of ITLPD-GI with inflammatory bowel disease (IBD), celiac disease, and autoimmune enteropathy [87].

### 5.2. Histopathology

ITCLD-GI is characterized by dense, non-destructive infiltration of small monotonous tumor cells into the lamina propria. Intraepithelial tumor cell infiltration is typically absent [5,79,84,85,87,101] (Figure 3). Tumor cell infiltration occasionally extends into the muscularis mucosae and submucosa, but mass formation or full-thickness involvement of the intestinal wall are generally not observed [85]. Tumor cells exhibit minimal atypia with small round nuclei, mature chromatin, and inconspicuous nucleoli [5,79,84,85,87,101]. A mixture of inflammatory cells is rare, but non-necrotizing epithelioid granulomas that resemble histopathology of Crohn’s disease may be present in a subset of cases [79,87].

### 5.3. Immunophenotype and Genetic Alterations

ITLPD-GI tumor cells typically exhibit a CD2^+^, CD3^+^, CD5^+^, and CD4^+^ or CD8^+^ phenotype [5,79,84,85,90,101]. Although uncommon, CD4^+^/CD8^+^ (double-positive) [90] and CD4^−^/CD8^−^ (double-negative) [85] phenotypes have been reported. In CD8^+^ cases, TIA-1 is positive, but granzyme B is typically negative [85,86,90]. CD56 [5,85,90] and EBV-encoded small RNA (EBER) in situ hybridization [5,85] are negative. CD30 [84,87,90,94] and CD103 [77,79,81,83,88] are also negative, with rare exceptions. Ki-67 labeling rate is very low (<10%) [5,84,85] (Figure 3). All cases exhibit a TCRβ (βF1)-positive phenotype [5,79,84,87,90,94]. All reported ITLPD-GI cases harbor clonal rearrangements of TR genes of TRG or TRB [84,85,87,90]. Although the molecular signatures underlying ITLPD-GI are poorly understood, targeted NGS recently revealed that the CD4^+^, CD4^+^/CD8^+^, and CD4^−^/CD8^−^ ITLPD-GI cases harbored frequent alterations of JAK-STAT pathway genes (5/6, 82%); *STAT3* SH2 domain hotspot mutations (D661Y and S614R) (*n* = 3, 50%), *SOCS1* deletion (*n* = 1, 16.7%), and *STAT3-JAK2* rearrangement (*n* = 1, 16.7%) [102]. Another study reported recurrent *STAT3-JAK2* fusions in 80% (4/5 cases) of CD4^+^ ITLPD-GIs, with evidence of STAT5 activation on immunostaining for pSTAT5^Y694^ [90].

### 5.4. Clinical Manifestations

ITLPD-GI can occur in any part of the GI tract, including the esophagus and oral cavity, but is more common in the small intestine and colon [79,84,85,87]. Patients present with diarrhea, abdominal pain, dyspepsia, vomiting and weight loss [79,84,85,87], and symptoms may be misdiagnosed as non-specific gastroenteritis, CD, or IBD. Extraintestinal expansion beyond the mesenteric lymph nodes is rare [83,84,85,86,87,88], but several cases involving the liver [79,87], bone marrow [83,86,87], and peripheral blood [87] have been reported. Endoscopically, ITLPD-GI exhibits thickening of the wall, irregular mucosa, and mucosal erosions. The surface of the mucosa is hyperemic with superficial erosion [79,84,85,87]. Multiple small polypoid lesions resembling lymphomatous polyposis are often reported [77,81,85].

### 5.5. Prognosis and Treatment Strategies

Progression is typically gradual, and patients have a chronic, refractory clinical course for years to decades. Although ITLPD-GI does not usually respond to conventional chemotherapy [79,84,85,87], steroids may improve symptoms [87]. Recently, a case of ITLPD-GI confined to the stomach treated with relatively low-dose (30 Gy) “involved-field radiation therapy” resulted in complete remission [97]. Although most patients have a relatively favorable prognosis with monitoring, a small percentage of patients demonstrate disease progression and transformation, typically after an interval of years, and cases with a CD4^+^ phenotype are considered to be at higher risk [79,84,90,94]. The vast majority of CD8^+^ ITLPD-GI are indolent disease, showing chronic course lasting for decades or spontaneous regression [103], transformation into a higher-grade lymphoma has also been reported in at least one case; Sharma et al. [90] reported a CD8^+^ ITLPD-GI that further had systemic ALK negative ALCL. The optimal treatment for this disease has yet to be established, and further case accumulation and long-term observations are needed. In the future, the use of targeted agents that directly affect the JAK-STAT pathway may be promising [104].

## 6. Differential Diagnoses and Diagnostic Pitfalls

### 6.1. Differential Diagnoses

Several aggressive T-cell and NK-cell lymphomas that typically arise outside the GI tract, including ALCL, ENKTCL, and adult T-cell leukemia/lymphoma (ATLL), may also involve the GI tract secondarily or primarily, although the latter is rare. NK-cell enteropathy is a non-progressive NK-cell proliferation and should be correctly differentiated from aggressive T-cell and NK-cell lymphoma. In addition, chronic active EBV infection (CAEBV) of the gastrointestinal tract may also be a differential diagnosis. In this section, we briefly describe these diseases and summarize points of differentiation.

#### 6.1.1. Anaplastic Large Cell Lymphoma (ALCL)

According to the current WHO classification, ALCL is divided into three types according to the presence or absence of ALK gene rearrangements and protein expression, and its primary site: systemic ALK-negative ALCL, systemic ALK-positive ALCL, and primary cutaneous ALCL [5]. Both ALK-negative and ALK-positive ALCLs involve lymph nodes and extranodal organs such as skin, soft tissue, liver, and lung. Extranodal sites other than skin is less frequently involved. The GI tract may be affected secondarily by systemic ALCL or may be the primary site. Primary GI ALCL is rare and has been reported in isolated case reports and small case series. In GI tract ALCL, ALK-negative ALCL is more prevalent [105,106], whereas ALK-positive ALCL is less prevalent [105,107,108,109,110] (approximately 24% of cases according to one report [105]).

Histologically, large pleomorphic cells proliferate diffusely. These large pleomorphic cells have irregular, and occasionally horseshoe- or kidney-shaped nuclei and abundant amphophilic cytoplasm (Figure 4). Lee et al. reported no angiodestruction or geographic necrosis and no evidence of enteropathies such as intraepithelial lymphocytes or crypt hyperplasia in background non-neoplastic mucosa [105]. Tumor cells are positive for CD30 in the Golgi and cytomembrane. In ALK-positive ALCL, the staining pattern varies depending on the translocation partner of ALK: some cases are positive in the nucleus and cytoplasm, whereas others are positive in the membrane and cytoplasm. Most cases are positive for at least one T-cell marker (CD2, CD3, CD4, CD5, and CD7). CD8 is typically negative, but most cases are positive for one of the cytotoxic markers (TIA-1, granzyme B, or perforin). OS is significantly better for ALK-positive ALCL than for ALK-negative ALCL [105,111,112].

#### 6.1.2. Extranodal NK/T Cell Lymphoma, Nasal Type (ENKTCL)

ENKTCL commonly arises in the nasal cavity or upper aerodigestive tract [5]. The GI tract may be affected secondarily as the disease progresses, but primary intestinal presentation has been reported in approximately 6% of cases [113,114,115].

Histologically, the size of the tumor cells is variable from small to large, and pleomorphic tumor cells are also present (Figure 5). The nuclei of the tumor cells are often folded and can be elongated, with granular chromatin and a moderate amount of pale to clear cytoplasm. An angiocentric/angiodestruction proliferation pattern and necrosis are often observed. Cases derived from NK cells are usually CD2^+^, CD3^+^ (cytoplasmic), CD5^−^, CD4^−^, CD8^−^, CD56^+^, and positive for TIA-1, granzyme B, and perforin. CD30 expression has been reported to vary but is observed in 26–47% of cases [116,117,118,119,120,121,122,123,124]. T-cell-derived tumor cells are positive for surface CD3^+^, CD5^+^, CD4^+^, or CD8^+^, or negative for both [113,114]. By definition, all cases demonstrate an association with EBV, and in situ hybridization for EBER demonstrates positive findings in the nuclei of many tumor cells [114]. Primary intestinal ENKTCL appears to have an inferior prognosis compared to ENKTCL originating in the upper aerodigestive tract [113,114].

#### 6.1.3. Adult T-Cell Leukemia/Lymphoma (ATLL)

ATLL is a rare T-cell neoplasm caused by chronic infection with the retrovirus human T-lymphotropic virus type 1 (HTLV-1) [5]. ATLL frequently involves the GI tract, especially the stomach. Ishibashi et al. reported that gastric, small intestinal, and large intestinal ATLL lesions were present in 66%, 18%, and 16% of cases, respectively [125]. ATLL of primary GI origin without leukemic changes or systemic lymphadenopathy has also been reported [126,127,128,129,130].

Histologically, medium to large, atypical lymphocytes proliferate diffusely. Primary gastric ATLL with lymphoepithelial lesions (LELs) have also been reported [129], and such cases may require attention for differentiation from extranodal marginal zone lymphoma of mucosa-associated lymphoid tissue (MALT lymphoma), EATL, or MEITL characterized by lymphocytic infiltration into the epithelium. Typically, tumor cells are CD2^+^, CD3^+^, CD4^+^, CD8^−^, CD5^+^, and CD25^+^, and CD7 is often negative (Figure 6). Large cells may be CD30^+^, which has been observed in 32% of cases of ATLL with GI tract involvement [125]. In such cases, differentiation from ALCL may be challenging, but ALK is negative. In addition, CD103 has been reported to be positive in 48% of ATLL with GI tract involvement [125].

#### 6.1.4. NK-Cell Enteropathy (NK-ENT)

NK-ENT, also known as lymphomatoid gastropathy, is a nonprogressive NK-cell proliferation that often mimics intestinal lymphoma as reported by Takeuchi et al. [131]. NK-ENT is asymptomatic and is not associated with a history of celiac disease, inflammatory bowel disease, or malabsorption. Many cases have been reported in patients infected with *H. pylori* or those with a history of gastric cancer, and regression has been observed in a subset of patients who received eradication therapy for *H. pylori*. However, a clear association between *H. pylori* infection and NK-ENT has yet to be established because NK-ENT regression is observed even in patients without *H. pylori* eradication, and a large proportion of Japanese patients are infected with *H. pylori* [131].

Histologically, medium-sized atypical lymphocytes diffusely infiltrate the lamina propria and occasionally infiltrate the glandular epithelium, resembling LEL. Proliferating cells typically have round and occasionally irregular nuclei, fine chromatin, inconspicuous nucleoli, and moderate to abundant clear or eosinophilic cytoplasm. Necrosis may be present, but angiodestruction is absent. Tumor cells typically exhibit a CD2^+/−^, cytoplasmic CD3^+^, CD4^−^, CD7^+^, CD8^−^, CD56^+^, CD103^−^, and TIA1^+^ phenotype (Figure 7). Rare CD8^+^ cases have also been reported [132]. Ki-67 labeling index varies among cases and is reported to be 10–90% [131,132,133,134,135]. EBER in situ hybridization is always negative, distinguishing this entity from ENKTCL. NK-ENT often regresses spontaneously without treatment, and malignant transformation has not been reported.

Recently, targeted NGS demonstrated recurrent *JAK3* mutations in 30% (3/10) of NK-ENT cases [135].

#### 6.1.5. Chronic Active Epstein-Barr Virus Infection (CAEBV)

CAEBV is usually observed in patients with congenital or acquired immunodeficiency [136,137]. CAEBV is one subtype of EBV-LPD and consists of a spectrum of lymphoid diseases including hyperplastic, borderline, and neoplastic diseases [5]. CAEBV has two types; T/NK-cell type and B-cell type, with the former being majority. Common symptoms include abdominal pain, fever, lymphadenopathy and splenomegaly, diarrhea, or weight loss. Endoscopically, diffuse granular, erythematous mucosa and multiple ulcers are common findings within the stomach or intestinal tract.

The histological findings vary depending on the grade of CAEBV, ranging from small to medium-sized lymphoid cells with minimal atypia that are difficult to distinguish from inflammatory bowel diseases to medium-sized lymphoid cells with mild to moderate atypia and abnormal T-cell marker expression. In all cases, numerous EBER-positive cells are observed [138]. Immunohistochemically, CD3^+^, CD8^+^, and CD4^−^ phenotypes have been reported in atypical cells. CD56 is also positive in T/NK-cell types [138].

Differentiating CAEBV from IBD is often challenging, as IBD may be superimposed with EBV infection. Extranodal NK/T-cell lymphoma should also be considered, as it shows similar histology and immunostaining results. CAEBV shows systemic symptoms and abnormal laboratory results from the onset, and the disease gradually worsens with repeated relapses, whereas NK/T cell lymphoma usually begins as a localized lesion, but cannot be reliably differentiated by itself. The diagnosis should be made carefully based on the clinical findings and blood test findings (e.g., EBV DNA copy number). Early detection of this disease is important because CAEBV can be associated with serious complications such as disseminated intravascular coagulation syndrome, gastrointestinal bleeding/perforation, myocarditis, and sepsis, and some patients with CAEBV die within a few years [138].

### 6.2. Diagnostic Pitfalls

#### 6.2.1. Discriminating Subtypes with Different Clinical Prognoses

The characteristics of T- and NK-cell lymphomas and lymphoproliferative diseases in the GI tract are summarized in Table 2. The basic immunostaining combinations that are recommended are CD20, CD3, CD5, CD4, CD8, CD30, CD56, EBER, and Ki-67, alongside additional stains such as CD2, CD7, CD103, TIA-1, granzyme B, perforin, MATK, CD79a, and PAX5, as necessary. The histopathological and immunostaining patterns presented in Table 2 are representative, and exceptions exist. The following pitfalls should be noted. The diagnosis should be based on a comprehensive evaluation of clinical information, including medical history (history of celiac disease or IBD), medication history (presence of immunosuppressive conditions), serological findings (HTLV-1, anti-tissue transglutaminase antibody and anti-endomysial antibody), and endoscopic findings.

The following points should be noted:Epitheliotropism is more clearly observed with cytokeratin (CK20) staining. Epitheliotropism is most frequently observed in EATL and MEITL but is also observed in other lymphomas or NK-ENT and is not specific to EATL and MEITL.EATL with large, atypical cell proliferation with CD30 expression is challenging to differentiate from ALK-negative ALCL, and a reliable differential approach has not been established.In MEITL, abnormal expression of CD20 is infrequently observed and may lead to misdiagnosis of B-cell lymphoma. In such cases, other B-cell markers such as CD79a and PAX5 should be evaluated to prevent misdiagnosis.In MEITL, CD8 or CD56 can be negative. CD56 negativity may cause difficulties in differentiating MEITL from EATL. In the case of CD8-negative MEITL, the immunostaining pattern may resemble that of NK-ENT, but the PCR results of NK-ENT do not exhibit TCR rearrangement.MATK is frequently expressed in MEITL, which may support the diagnosis, but is also reported to be positive in up to 87% of ENKTCL cases [61,114], which should be noted.CD103 expression may represent neoplastic changes in mucosal intraepithelial lymphocytes, but it should be noted that approximately half of ATLL cases are also positive for CD103.Despite NK-ENT being clinically indolent, Ki-67 labeling index has been reported to be high in some cases. In this situation, high rate of Ki-67 LI is not necessarily an indicator of aggressive clinical course.NK-ENT can be misdiagnosed as ENKTCL, but EBER in situ hybridization will always show negative results for the former and positive results for the latter.

#### 6.2.2. Distinguishing Primary GI Neoplasms from Secondary Involvement

Detailed history-taking and investigation of systemic lymphadenopathy, paranasal sinus lesions, skin lesions, and bone marrow lesions are crucial when considering the primary site of the disease. Imaging techniques such as computed tomography (CT) and positron emission tomography (PET) scans are also useful for evaluating the distribution of lesions. When no lymphadenopathy or organ involvement other than the intestinal tract are noted, the entity is more likely to be a primary GI T- or NK- cell lymphoma. However, in advanced primary GI tract lymphoma cases, diagnosis is difficult because the lesion extends outside the GI tract.

#### 6.2.3. Differentiating Neoplasms from Non-Specific Inflammatory Changes at an Early Stage

In small biopsy specimens, aggressive GI T- and NK- cell lymphoma can be misdiagnosed as inflammatory disorders such as celiac disease IBD, lymphocytic colitis, and autoimmune enteropathy [139]. In particular, when the distant IEL zone of a MEITL lesion is sampled, it may be challenging to recognize the entity as a lymphoma. In the IEL zone of MEITL, normal-appearing IEL is positive for CD8 and negative or weakly positive for CD56 and MATK [58]. It is also necessary to recognize abnormal IEL findings (>20 IELs per 100 epithelial cells [140]) to diagnose IBD, celiac disease, lymphocytic colitis, EATL, and MEITL. The IEL is denser in lymphoma with IEL than in inflammatory disorders. Accordingly, it is crucial to perform a diagnosis by considering endoscopic findings and medical history.

Differentiating ITLPD-GI from these inflammatory disorders can be more challenging, as the mucosa may exhibit similar architectural changes and T-cell antigen abnormalities may not be present. Moreover, occasional epithelioid granulomas have been reported in patients with ITLPD-GI, which can be misdiagnosed as Crohn’s disease [79,83,141]. In ITLPD-GI, the lack of a variety of inflammatory cells and the monotonous proliferation of small lymphoid cells may lead to the recognition of this disease. Endoscopic and serological findings, immunophenotype, and genetic analysis are helpful, but a high clinical suspicion is essential to diagnose this subtype of lymphoma.

## 7. Conclusions

Primary GI T-cell lymphomas include various diseases such as EATL and MEITL, which have an extremely poor prognosis. In contrast, ITLPD-GI progresses slowly and is often over-treated. A comprehensive framework that integrates clinical manifestations, histology, immunostaining results, and genetic data is required for accurate diagnosis. For any entity included in this group, the optimal treatment strategy needs to be established using further accumulation of data.

## Figures and Tables

**Figure 1 cancers-13-05774-f001:**
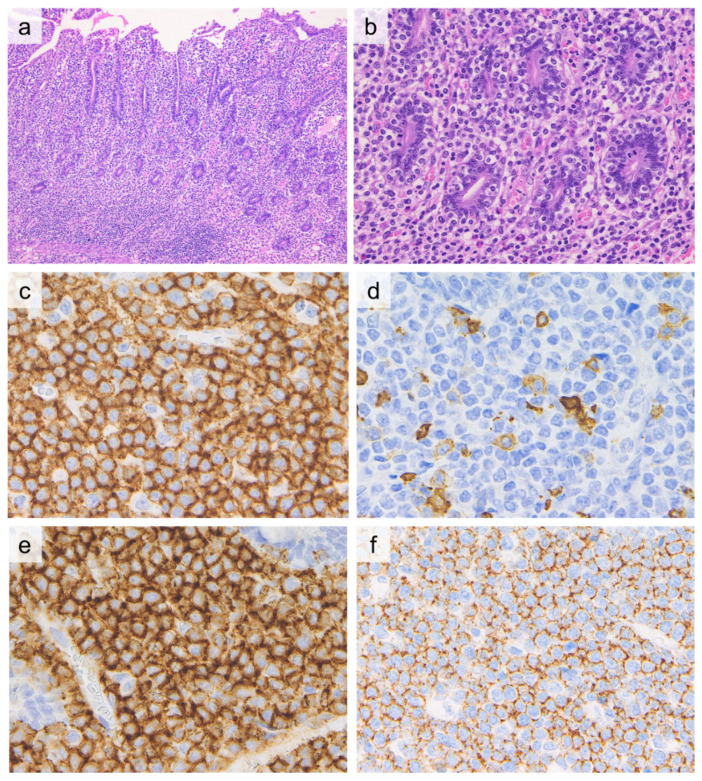
Histological and immunohistochemical findings of monomorphic epitheliotropic intestinal T-cell lymphoma. ((**a**, 10×, **b**, 40×), H&E) Diffuse proliferation of monomorphic lymphoid cells with severe infiltration into the crypt epithelium is observed. Tumor cells are CD3-positive (**c**, 60×), CD5-negative (**d**, 60×), CD8-positive (**e**, 60×), and CD56-positive (**f**, 60×).

**Figure 2 cancers-13-05774-f002:**
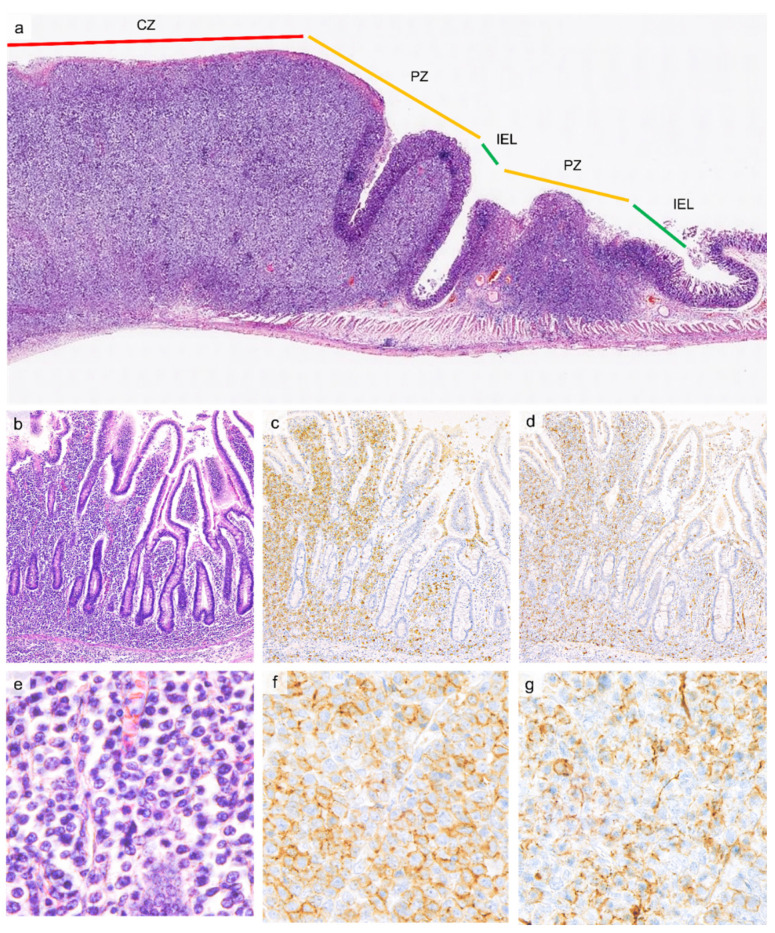
((**a**, low-power view), H&E) Jejunal resection demonstrating the central zone (CZ) characterized by transmural infiltration by lymphoma, the peripheral zone (PZ) characterized by lateral spread of the lymphoma into the mucosa and submucosa, and the distant intraepithelial lymphocytosis zone. ((**b**, 20×), H&E) The border area between PZ and intraepithelial lymphocytosis zone is shown. In intraepithelial lymphocytosis zone, IELs show minimal atypia ((**e**, 60×), H&E) and are positive for CD8 (**c**, 20×, **f**, 60×) and CD56 (**d**, 20×, **g**, 60×).

**Figure 3 cancers-13-05774-f003:**
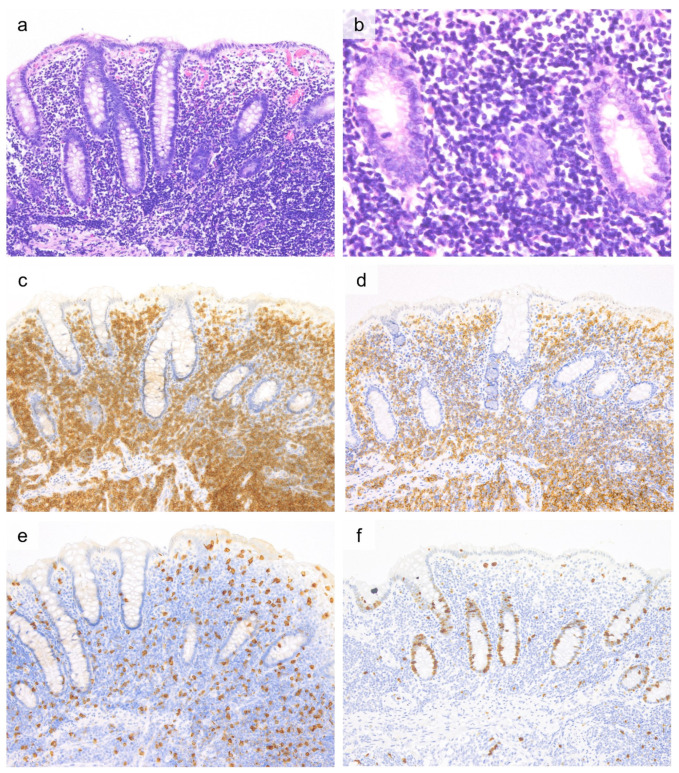
Histological and immunohistochemical findings of indolent T-cell lymphoproliferative disorder of the gastrointestinal tract. ((**a**, 20×), H&E) Cecal biopsy shows dense lymphoid cell infiltration in lamina propria and muscularis mucosae, and the mucosal architecture is relatively preserved. ((**b**, 40×), H&E) The lymphoid cells are small to medium-sized with minimal atypia. These lymphoid cells are CD3 positive (**c**, 20×), CD4 positive (**d**, 20×), and CD8 negative (**e**, 20×). Ki-67 labeling index is low (<5%) (**f**, 20×).

**Figure 4 cancers-13-05774-f004:**
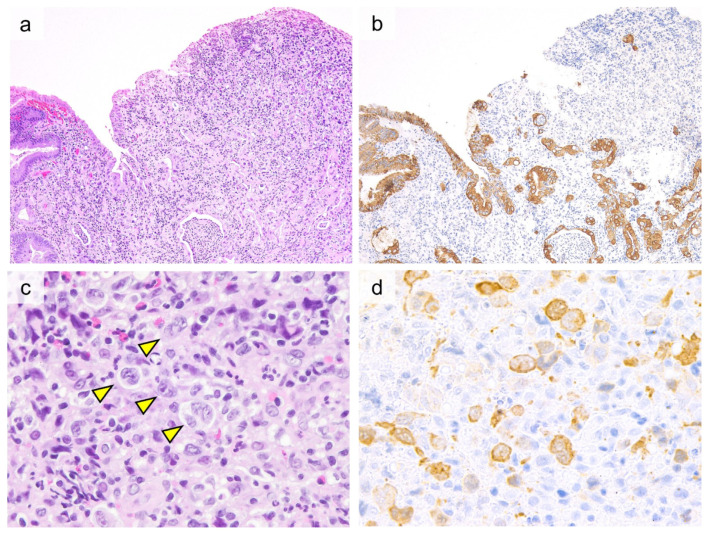
Histological and immunohistochemical findings of anaplastic lymphoma kinase (ALK)-positive anaplastic large cell lymphoma involving the gastric mucosa. ((**a**, 20×), H&E) Tumor cells are proliferating diffusely in the gastric mucosa. ((**b**, 20×), Cytokeratin AE1/AE3) Cytokeratin staining highlights the destruction of the glandular epithelium. ((**c**, 60×), H&E) “Hallmark” cells ((indicated by an arrowhead) with horseshoe- or kidney-shaped nuclei are easily found. ((**d**, 60×), ALK) Tumor cells showed cytoplasmic staining pattern for ALK.

**Figure 5 cancers-13-05774-f005:**
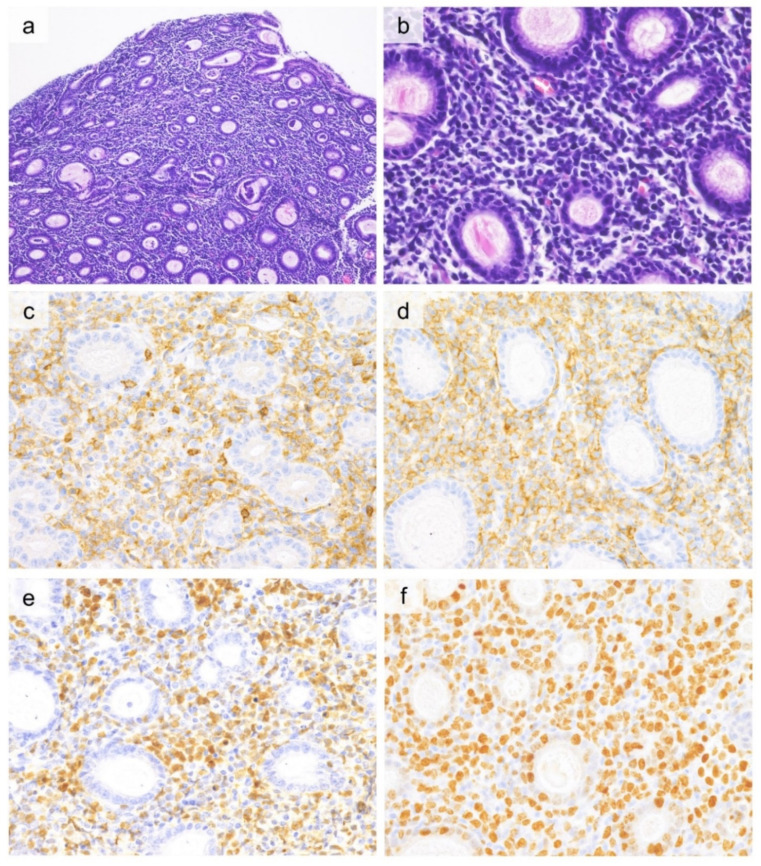
Histological and immunohistochemical findings of extranodal NK/T cell lymphoma, nasal type. ((**a**, 20×, **b**, 40×), H&E) Dense proliferation of small to medium-sized lymphocytes in the gastric mucosa. These lymphoid cells are positive for CD8 (**c**, 40×) and CD56 (**d**, 40×). Most tumor cells are positive for EBER in situ hybridization (**e**, 40×). Ki-67 labeling index is high (**f**, 40×).

**Figure 6 cancers-13-05774-f006:**
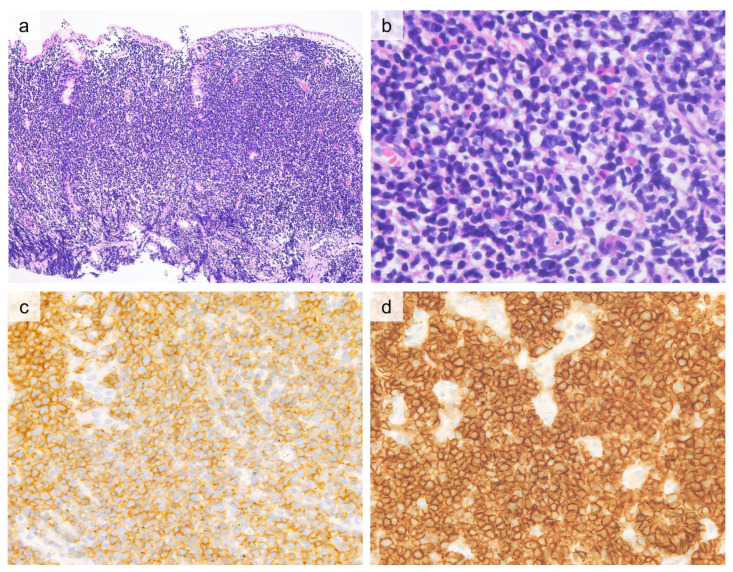
Histological and immunohistochemical findings of adult T-cell leukemia/lymphoma. ((**a**, 20×, **b**, 40×), H&E) Gastric biopsy shows dense infiltration of medium to large-sized atypical lymphocytes. In high magnification view, tumor cells occasionally show nuclear indentation. Tumor cells are positive for CD4 (**c**, 40×) and CD25 (**d**, 40×).

**Figure 7 cancers-13-05774-f007:**
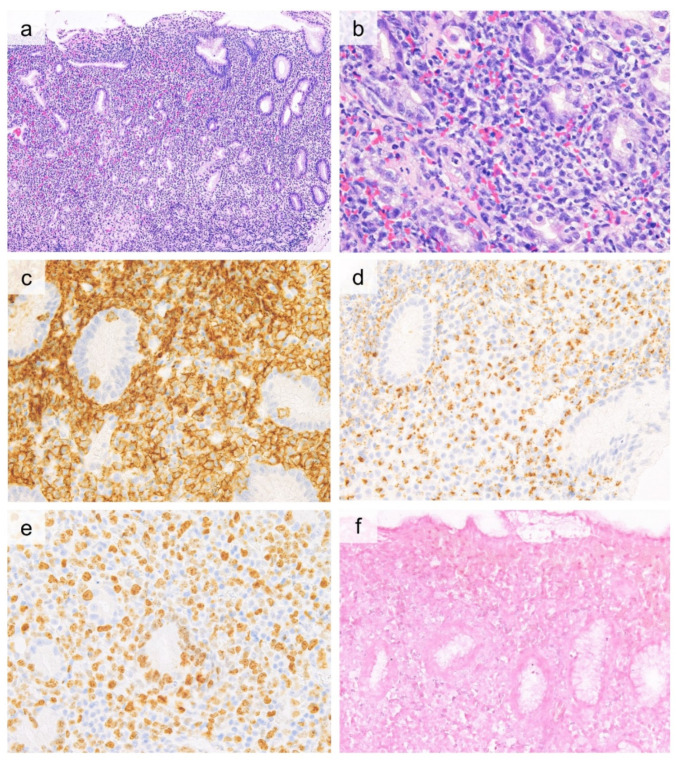
Histological and immunohistochemical findings of NK-cell enteropathy. ((**a**, 20×, **b**, 40×), H&E) Medium-sized atypical lymphocytes diffusely infiltrate the gastric mucosa. Tumor cells have irregular nuclei. Tumor cells are positive for CD56 (**c**, 40×) and TIA-1 (**d**, 40×). Ki-67 labeling index is relatively high (approximately 50%) in this case (**e**, 40×). EBER in situ hybridization is negative (**f**, 40×).

**Table 1 cancers-13-05774-t001:** Comparison of celiac disease, RCD I, RCD II, and EATL [5,9,19,20,21,22,23,24,25].

Investigations	Celiac Disease	RCD I	RCD II	EATL
Disease type	Chronic enteropathy triggered by dietary gluten	Persistent autoinflammatory immune response, gluten independent	Low-grade lymphoproliferative disorder	High-grade lymphoma
Immunophenotype	sCD3^+^, cCD3^+^, CD5^−/+^, CD8^+^, CD103^+^	sCD3^+^, cCD3^+^, CD5^−/+^, CD8^+^, CD103^+^	sCD3^−^, cCD3^+^, CD5^−^, CD8^−^, CD103^+^, CD30^−^	CD3^+^, CD8^−^, CD30^+^, Ki67 LI: high (>50%)
T-cell receptor	polyclonal	polyclonal	monoclonal	monoclonal
5-year survival		80–96% [18,20,21]	45–58% [18,20,21]	~20% [5,9,20]
Rate of progression to EATL in 5 years	0.7% [24]	3–14% [19,20,21,22,23]	33–52% [19,20,21,22,23]	-

Abbreviations: RCD; refractory celiac disease, EATL; enteropathy-associated T-cell lymphoma, sCD3; surface CD, cCD3; cytoplasmic CD, Ki-67 LI; Ki-67 labeling index.

**Table 2 cancers-13-05774-t002:** Comparison of the characteristics of gastrointestinal NK- and T-cell lymphoma/lympho-proliferative diseases.

Investigation	EATL	MEITL	ALCL	ENKTCL	ATLL	ITLPD-GI	NK-ENT
Common sites	Jejunum Ileum	Jejunum Ileum	Small intestine Stomach Colon Esophagus	Small intestine Colon Ileocecal junction	Stomach Small intestine Colorectal	Small intestine Colon Esophagus Oral cavity	Stomach Duodenum Small intestine Colon
Morphology	- Pleomorphic - Medium to large-sized cells - Necrosis - Reactive inflammatory cell infiltration - Epitheliotropic	- Monomorphic - Small to medium sized cells - Epitheliotropic	- Large pleomorphic cells - Horseshoe- or kidney-shaped nuclei	- Small to large sized cells - Pleomorphic cell admixture - Folded or elongated cell - Angiocentric pattern - Necrosis	- Medium to large-sized atypical cells - Indented and lobulated nuclei	- Small monotonous cells - Minimal atypia - Occasional epithelioid granulomas	- Medium-sized cells - Sometimes epitheliotropic - No angiocentric pattern
Immuno- phenotype							
CD2	+	+	Most cases are positive for at least one of the T-cell markers	+	+	+	−/+
CD3	+	+		+	+	+	+ (cytoplasmic)
CD5	−	−		−/+	+	+	−
CD4	−	−		−	+	CD4^+^ or CD8^+^	−
CD8	−/+	+	−	−/+	−		−
CD30	frequently + in large cells	−	+ (in Golgi and cytomembrane)	sometimes + (26–47%)	sometimes + (32%)	−	−
CD56	−	+	− sometimes+ (~20%)	+	−	−	+
TIA-1	+	+	+ (+for at least one of the cytotoxic markers)	+	−	+ in CD8^+^ cells	+
EBER in situ	−	−	−	+	−	−	−
Other findings	CD103^+^, MATK^+^ < 40%	CD103^+^, MATK^+^ > 85%, Aberrant CD20^+^ (11–24%)	ALK^+^ for approximately 24% in GI-ALCL	Cases derived from NK cells are sCD3^−^ and cCD3^+^	CD25^+^ CD103^+^ (48%)	Ki-67 LI is very low (<10%)	CD103^−^ Ki-67 LI varies in the range of 10–90%
Chromosomal features	Gains of 9q, 7q, 1q, 5q Losses of 6q	Gains of 9q, 8q	(ALK^+^ cases) Gains of 17p, 17q, 7p Losses of 4q, 11q (ALK^−^ cases) Gains of 6p, 7p	Gains of 1q, 17q, 20q Losses of 6q, 8p, 11q	Gains of 14q Losses of 6q	−	−
Genetic features	JAK-STAT RAS	JAK-STAT RAS SETD2	(ALK^+^ cases) NPM-ALK fusion JAK/STAT RAS-ERK PIK-AKT	JAK-STAT Epigenetic regulators	TCR-NF-κB JAK/STAT	(CD4^+^ cases) JAK/STAT RAS Epigenetic modifiers	JAK/STAT
Clinical course	Aggressive	Aggressive	Aggressive	Aggressive	Aggressive	Indolent, Slight risk of progression and transformation	Indolent, No transformation reported

Abbreviations: EATL; enteropathy-associated T-cell lymphoma, MEITL; monomorphic epitheliotropic intestinal T-cell lymphoma, ALCL; anaplastic large cell lymphoma, ENKTCL; extranodal NK/T cell lymphoma, nasal type, ATLL; adult T-cell leukemia/lymphoma, ITLPD-GI; indolent T-cell lymphoproliferative disorder of the gastrointestinal tract, NK-ENT; NK-cell enteropathy, Ki-67 LI; Ki-67 labeling index.

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
