# Peer review of "Primary Gastrointestinal T-Cell Lymphoma and Indolent Lymphoproliferative Disorders: Practical Diagnostic and Treatment Approaches"

_cancers, 2021, doi:10.3390/cancers13225774_

Round 1
Reviewer 1 Report
In this review, Nishimura et al cover the clinicopathologic and genetic features of intestinal T-cell lymphoproliferative disorders, and discuss treatment considerations. The review highlights many key issues; however, the manuscript could be strengthened by including additional recent discoveries, particularly regarding newer molecular genetic findings in these entities. The following are recommendations that would help strengthen this review manuscript.
- The authors describe RCDII as a precursor state to (some forms of) EATL. Recent studies have profiled molecular alterations in RCDII and shown substantial overlap with those seen in EATL, including recurrent mutations in the JAK-STAT pathway, NF-kB pathway, epigenetic modifiers, amongst others. It would strengthen the manuscript to include this updated molecular data for RCDII.
- The immunophenotype of CD and RCDI are all described as CD5-. It is true that the IELs in CD and RCDI may show variable expression (or downregulation) of CD5, however, they are generally not entirely CD5 negative. In some cases, there may be subsets which are positive and negative, therefore it may be better to describe the immunophenotype of CD and RCDI as “CD5-/+”.
- Though substantially less common, some cases of RCDII may be CD8+. This could also be included in the text (though Table 2 could still be kept as “CD8-”, since the majority of cases are CD8 negative).
- Since this manuscript includes a discussion of treatment strategies, it may be reasonable to include a few sentences on treatment of RCDII, particularly the use of budesonide.
- In section 2.5, “Surface TTCR” should be corrected to “Surface TCR.”
- In section 3.5, “Standardized treatment strategies for EATL have…” should be changed to “Standardized treatment strategies for MEITL have…”
- In addition to the STAT3-JAK2 rearrangement, recent studies have identified additional molecular alterations in cases of indolent T-cell lymphoproliferative disorders, including mutations in STAT3, amongst others. It would strengthen the manuscript to include this updated molecular data.
- In section 5.5, “ITCLD-GI” should be corrected to “ITLPD-GI”
- It is true that most cases of transformed ITLPD-GI are CD4+, however, at least one CD8+ case is also known to have transformed.
- Recent studies have identified recurrent molecular alterations in JAK3 in NK-cell enteropathy. It would strengthen the manuscript to include this updated molecular data.
- In section 6.2.1, regarding NK-ENT in the diagnostic pitfalls, “high rate of Ki-67 LI is not necessarily an indicator of malignancy” could potentially be changed to “high rate of Ki-67 LI is not necessarily an indicator of aggressive clinical course” to avoid confusion about the neoplastic nature of the entity.
- In section 6.2.3, the abbreviation for “ITLPD-GI” has been incorrectly written twice (as “ITLPD-G”, and “ITLDPD-GI”).
Author Response
Comment from Reviewer 1
・Comment 1: [The authors describe RCDII as a precursor state to (some forms of) EATL. Recent studies have profiled molecular alterations in RCDII and shown substantial overlap with those seen in EATL, including recurrent mutations in the JAK-STAT pathway, NF-kB pathway, epigenetic modifiers, amongst others. It would strengthen the manuscript to include this updated molecular data for RCDII.]
Response 1: Thank you very much for reviewing our manuscript and providing important information. As you suggested, we have added the information about molecular alternations in RCDII in the manuscript and stated about overlap with EATL. (Page 4, Line 171-176)
・Comment 2: [The immunophenotype of CD and RCDI are all described as CD5-. It is true that the IELs in CD and RCDI may show variable expression (or downregulation) of CD5, however, they are generally not entirely CD5 negative. In some cases, there may be subsets which are positive and negative, therefore it may be better to describe the immunophenotype of CD and RCDI as “CD5-/+”.]
Response 2: As you pointed out, we have changed the description in the manuscript and Table 1.
(Page 3, Line 115, 117-119 and Table 1.)
・Comment 3:[Though substantially less common, some cases of RCDII may be CD8+. This could also be included in the text (though Table 2 could still be kept as “CD8-”, since the majority of cases are CD8 negative).]
Response 3: Unfortunately, we couldn’t find any literature that reported CD8 positivity rate in RCD II, but we did find a report that demonstrated monoclonality of the TCR gene among patients who met the clinical criteria for RCD and were CD8 positive, so we included it in the manuscript.
(Page 3, Line 119-122).
・Comment 4: [Since this manuscript includes a discussion of treatment strategies, it may be reasonable to include a few sentences on treatment of RCDII, particularly the use of budesonide.]
Response 4: As you suggested, I mentioned about the treatment of RCD II, focusing on the use of budesonide. (Page 5, Line 216-224.)
・Comment 5:[ In section 2.5, “Surface TTCR” should be corrected to “Surface TCR.”]
Response 5: The typo you pointed out has been corrected. (Page 4, Line 160)
・Comment 6:[ In section 3.5, “Standardized treatment strategies for EATL have…” should be changed to “Standardized treatment strategies for MEITL have…”]
Response 6: We have corrected the part you pointed out. (Page 8, Line 297)
・Comment 7:[ In addition to the STAT3-JAK2 rearrangement, recent studies have identified additional molecular alterations in cases of indolent T-cell lymphoproliferative disorders, including mutations in STAT3, amongst others. It would strengthen the manuscript to include this updated molecular data.]
Response 7: Thank you for your valuable comments. We have added these recent findings on molecular alternations obtained by targeted next-generation sequencing to the manuscript.
(Page 10, Line 378-381)
・Comment 8:[ In section 5.5, “ITCLD-GI” should be corrected to “ITLPD-GI”]
Response 8: The typo you pointed out has been corrected. (Page 12, Line 405)
・Comment 9:[ It is true that most cases of transformed ITLPD-GI are CD4+, however, at least one CD8+ case is also known to have transformed.]
Response 9: As you suggested, we have described about at least one case of CD8+ ITLPD-GI that transformed into a higher-grade lymphoma. (Page 12, Line 411-415)
・Comment 10:[ Recent studies have identified recurrent molecular alterations in JAK3 in NK-cell enteropathy. It would strengthen the manuscript to include this updated molecular data.]
Response 10: As you suggested, we have added molecular data of NK-cell enteropathy to the manuscript. (Page 16, Line 521-522)
・Comment 11:[ In section 6.2.1, regarding NK-ENT in the diagnostic pitfalls, “high rate of Ki-67 LI is not necessarily an indicator of malignancy” could potentially be changed to “high rate of Ki-67 LI is not necessarily an indicator of aggressive clinical course” to avoid confusion about the neoplastic nature of the entity.]
Response 11: We have corrected the manuscript as you suggested. (Page 20, Line 599-600)
・Comment 12:[ In section 6.2.3, the abbreviation for “ITLPD-GI” has been incorrectly written twice (as “ITLPD-G”, and “ITLDPD-GI”).]
Response 12: The two typos you pointed out has been corrected. (Page 21, Line 628, 629)
Reviewer 2 Report
The review is well-written and gives a comprehensive overview on the complicated topic of T/NK-cell lymphomas of the gastrointestinal tract.
I have some considerations:
- I would mention in the paper CAEBV as this entity is often difficult to be diagnosed in particular on small biopsy samples and enters in the differential diagnosis with both benign and malignant conditions discussed in the paper.
- I would add the following reference "Indolent T-cell lymphoproliferative disorders of the gastrointestinal tract (ITLPD-GI): a review" Cancers 2021,13( 11): 2790
Minor points: - Table 1 is correctly named in the text, whereas in the legend it is named as table 2.
- In Table 1 legend "comparison of between" need to be corrected
- Line 173 "common sites of involution" instead of involvement
- Line 272: treatment strategies for EATL(I believe treatment strategies for MEITL should be the correct sentence)
- I would add magnification in all figures and I would specify if the images are original from the Authors
- In Figures 2c and 2d I would use a higher magnification to highlight IELs
Author Response
Comment from Reviewer 2
・Comment 1: [I would mention in the paper CAEBV as this entity is often difficult to be diagnosed in particular on small biopsy samples and enters in the differential diagnosis with both benign and malignant conditions discussed in the paper.]
Response 1: Thank you for reviewing our manuscript. We have discussed the diseases you mentioned in section 6.1.5. (Page 18, Line 527-552)
・Comment 2: [I would add the following reference "Indolent T-cell lymphoproliferative disorders of the gastrointestinal tract (ITLPD-GI): a review" Cancers 2021,13( 11): 2790]
Response 2: We have cited the paper you suggested in Section 5.1. (Reference number #100)
・Comment 3: [Table 1 is correctly named in the text, whereas in the legend it is named as table 2. ]
Response 3: The title of the table has been corrected from Table2 to Table1. (Page 3, Line 125)
・Comment 4: [In Table 1 legend "comparison of between" need to be corrected.]
Response 4: We have corrected the legend of Table 1. (Page 3, Line 125)
・Comment 5: [Line 173 "common sites of involution" instead of involvement]
Response 5: We have appropriately corrected the part that you pointed out. (Page 5, Line 187-188)
・Comment 6: [Line 272: treatment strategies for EATL(I believe treatment strategies for MEITL should be the correct sentence)]
Response 6: We have corrected the part that you pointed out. (Page 8, Line 297)
・Comment 7: [I would add magnification in all figures and I would specify if the images are original from the Authors]
Response 7: Thank you for your suggestions. I have considered your suggestion to add magnifications in all the figures with my colleagues, but I'm afraid to say that we are unable to accept it. It is not common in pathology journals to include magnification in the figure legend because the magnification of histological images changes in the process of cropping and scaling. Therefore, we could not add the magnification in each legend. All histological photographs are our original. We could not include the statement about figure originality because there is no section in the manuscript to specify it. If you have an idea for a section suitable for a statement on originality, please let us know.
・Comment 8: [In Figures 2c and 2d I would use a higher magnification to highlight IELs]
Response 8: Figure 2, a-c, has been replaced with higher magnification images, and we have added further higher magnification pictures to highlight the IELs (Figure 2, e-g).
Reviewer 3 Report
I read the review about gastrointestinal T-cell lymphoma with great interest. It is well described, comprehensive, and worth for publication. I have only one request to the authors. Please provide chromosomal and mutational features of each type of T-cell lymphoma in the table 2.
Author Response
Comment from Reviewer 3
・Comment 1: [I read the review about gastrointestinal T-cell lymphoma with great interest. It is well described, comprehensive, and worth for publication. I have only one request to the authors. Please provide chromosomal and mutational features of each type of T-cell lymphoma in the table 2.]
Response 1: Thank you for reviewing our manuscript and for an important advice. We have provided the chromosomal and mutational features of each type of T-cell lymphoma in table 2.